# Theoretical Investigation on the Hydrogen Evolution, Oxygen Evolution, and Oxygen Reduction Reactions Performances of Two-Dimensional Metal-Organic Frameworks Fe_3_(C_2_X)_12_ (X = NH, O, S)

**DOI:** 10.3390/molecules27051528

**Published:** 2022-02-24

**Authors:** Xiaohang Yang, Zhen Feng, Zhanyong Guo

**Affiliations:** 1School of Science, Henan Institute of Technology, Xinxiang 453000, China; yangxh@hait.edu.cn; 2School of Materials Science and Engineering, Henan Institute of Technology, Xinxiang 453000, China; guozhanyong123@126.com; 3School of Physics, Henan Normal University, Xinxiang 453007, China

**Keywords:** two-dimensional metal-organic framework, ligand, single-atom catalysts, hydrogen evolution reaction, oxygen evolution reaction, oxygen reduction reaction

## Abstract

Two-dimensional metal-organic frameworks (2D MOFs) inherently consisting of metal entities and ligands are promising single-atom catalysts (SACs) for electrocatalytic chemical reactions. Three 2D Fe-MOFs with NH, O, and S ligands were designed using density functional theory calculations, and their feasibility as SACs for hydrogen evolution reaction (HER), oxygen evolution reaction (OER), and oxygen reduction reaction (ORR) was investigated. The NH, O, and S ligands can be used to control electronic structures and catalysis performance in 2D Fe-MOF monolayers by tuning charge redistribution. The results confirm the Sabatier principle, which states that an ideal catalyst should provide reasonable adsorption energies for all reaction species. The 2D Fe-MOF nanomaterials may render highly-efficient HER, OER, and ORR by tuning the ligands. Therefore, we believe that this study will serve as a guide for developing of 2D MOF-based SACs for water splitting, fuel cells, and metal-air batteries.

## 1. Introduction

The greenhouse effect, air pollution, ozone depletion, and fossil fuel depletion are all major challenges for our society’s progress in the 21st century [1]. To address the environmental deterioration and energy challenges, it has become a major priority to increase the research and development of low-cost, efficient, and renewable energy storage and conversion devices, such as fuel cells, metal-air cells, and water decomposition [2]. At the United Nations Climate Summit 66 countries pledged to achieve net-zero carbon emissions by 2050 [3]. A promising energy conversion technology is the unitized regenerative fuel cell. It works like a fuel cell and inversely as a water electrolyzer to produce H_2_ and O_2_ to feed the fuel cell. Hence, multifunctional electrocatalysts play key roles [4]. However, because of the high overpotential, low activity, and poor selectivity, it is extremely desirable to develop sustainable and low-cost functional electrode materials with high energy density, excellent rate capability, and good cycling stability [5]. 

Since an isolated Pt single atom anchored in FeO_x_ showed remarkable catalytic performance for CO oxidation [6], single-atom catalysts (SACs) have been considered next-generation electrode candidates. SACs contain isolated single-metal atoms dispersed on a support surface, and represent the ultimate limit of atom use efficiency for catalysis [7]. Some experimental and computational studies show that SACs are promising for precise control of catalytic reactions, such as the hydrogen evolution reaction (HER) [8,9], the oxygen evolution reaction (OER) [10,11], the oxygen reduction reaction (ORR) [12], the nitrogen reduction reaction (NRR) [13], the carbon dioxide reduction reaction (CO_2_RR) [14], and CO/NO oxidation [15]. 

Unlike typical SACs, two-dimensional metal-organic frameworks (2D MOF) contain metal entities and organic ligands, indicating that they could be used as SACs [16]. MOF monolayers have highly exposed metal atoms, uniformly dispersed against agglomeration [16]. Similar to SACs, the metal entities in MOFs could effectively modify charge redistribution and boost chemical reactions [17]. 

Recently, more and more 2D MOF sheets have been experimentally synthesized and theoretically predicted for use as catalysts [18]. The Cu_3_(C_6_S_6_) MOF cathode enables a high reversible capacity for lithium-ion batteries [19]. The Rh_3_C_12_S_12_ MOF exhibits a low limiting potential of –0.43 V for CO_2_RR [20]. Mo_3_C_12_N_12_H_12_ MOF exhibits a low overpotential of 0.18 V for NRR [21]. Mo_3_(C_2_O)_12_ MOF could achieve a low limiting potential of –0.36 V for NRR via the distal pathway [22]. These MOFs could also be used as electrocatalysts for ORR [23], OER [24], and HER [25]. 

Due to the above results, the NH, O, and S ligands were adopted to design 2D Fe-MOFs. The potential as SACs for the HER, OER, and ORR was systematically explored. The Fe-MOF exhibits atomically thin like 2D graphene, and they display excellent structural stability. The NH, O, and S ligands could tune charge redistribution in Fe-MOF and catalysis performance. The Fe-O MOF displays Δ*G*_H_ = 0.08 eV for HER, Fe-NH MOF exhibits *η*^ORR^ = 0.38 V for ORR, and they possess poor OER catalysis performance (*η*^OER^ > 0.92 V). Our work highlights the effect of ligands, and could guide the development of highly effective SACs based on 2D MOFs. 

## 2. Results and Discussion

### 2.1. Geometry and Stability

The unit cells of the studied 2D Fe-MOF monolayers are depicted in Figure 1. There are three types of ligating atoms between Fe atoms and graphene nanosheets (Appendix A). Therefore, different symbols were denoted by different ligating atoms: Fe-NH-MOF for NH ligating atoms, Fe-O-MOF for O ligating atoms, and Fe-S-MOF for S ligating atoms, respectively. These 2D MOF sheets are also atomically thin like grapheme, but each unit cell consists of 3 Fe atoms, 24 carbon atoms, and 12 ligating atoms. 

To optimize the atomic structures of these three Fe-MOFs, the variations of energies vs. the lattice constants are also shown in Figure 1, their lattice constants are optimized to be 12.61 Å for Fe-NH MOF, 12.31 Å for Fe-O MOF, and 13.65 Å for Fe-S MOF, respectively (Table 1 and Appendix A). These optimized lattice constants agree with previous investigations [22,26,27]. The bond lengths of Fe-N, Fe-O, and Fe-S are 1.85 Å, 1.83 Å, and 2.15 Å, respectively, and the bond lengths of C-N, C-O, and C-S are 1.35 Å, 1.30 Å, and 1.74 Å, respectively, due to differences in the atomic radius of N (r = 70 pm), O (r = 66 pm), and S (r = 104 pm). The diameters of the holes in Fe-MOF sheets vary as well by 3.44 for Fe-NH MOF, 5.17 Å for Fe-O MOF, and 5.74 Å for Fe-S MOF, respectively (Table 1). 

We examined the stability of three Fe-MOF monolayers after attaining their unique structures, because the good stability of the given materials is a prerequisite for their practical uses. Notably, high-quality 2D Fe-S MOF have been experimentally synthesized [27], however, their thermal stabilities were still performed using first-principles plus ab initio molecular dynamics simulations (AIMD). At a temperature of 500 K, a total time process of 3000 fs with a time step of 1 fs was implemented using specified 2 × 1 × 1 rectangular supercells (included 102 atoms for Fe-NH MOF and 78 atoms for Fe-O/ Fe-S MOFs). The variations of total energy and temperature and the final snapshots in 3000 fs are depicted in Figure 2. Their total energies and temperature exhibit up and down trends within a fixed range. These final structures display lack of structural distortion and no bond-breaking. It slight up and down changes in final plane structure can be seen. These results revealed that these three Fe-MOF monolayers could maintain their original atomic structures at a high temperature of 500 K, implying their exceptional thermal stability. The high stability may result from the large π-bonds of high-symmetric sp^2^-C atoms in graphene nanosheets, which agrees with previous work [22,27]. Notable are the slight up and down changes in the plane. The plane structure shows some fluctuation changes.

We further perform Bader charge analysis to investigate the chemical bonding in these Fe-MOF monolayers. The electron localization function (ELF) map and isosurfaces of ELF with a value of 0.50 au are plotted in Figure 3a–c, the Fe loses 0.58–1.51 *e*, and the N, O, and S gain 0.13–1.07 *e*, which contributes to their robust ionic bonds. Note that the positive Fe atom may be used as an active site for chemical reactions. 

### 2.2. Electronic Property

Previous research has shown that the electronic structures of 2D-based catalysts have a significant impact on their catalytic efficiency [28]. Thus, we computed their band structures and density of states (DOS) with the DFT + U method [29]. As shown in Figure 4a, the Fe-O MOF and Fe-S MOF display intrinsic metallicity due to the several bands at the Fermi level, while Fe-NH MOF is a semiconductor with band gaps of 0.56 eV for spin up and 0.89 eV for spin down. Thus, the high electrical conductivity of Fe-O and Fe-S MOFs should ensure rapid charge transfer in electrochemical reactions. It can be clearly observed that all three Fe-MOF nanomaterials possess spin splitting of band structures, producing magnetism. The computed total magnetic moment (*M*_tot_) of the primitive cells of Fe-NH-MOF, Fe-O-MOF, and Fe-S-MOF monolayers are 6.00 μB, 10.45 μB, and 9.25 μB, respectively. Further study of magnetism comes from the spin-charge density in Figure 3(a3,b3,c3), the spin-up densities are mainly around the Fe atoms, matching their total magnetic moments (Table 1).

Previous research suggested that metallic characteristics and strongly spin-polarized Fe atoms could enhance the chemical catalysis process [30]. The density of states (DOS) of these three Fe-MOFs was then calculated to further understand them better. The projected density of states (PDOS) of Fe, N, O, S, and C elements are further plotted in Figure 4b–d. There are obvious hybridizations between Cu-dyz and Cu-dxz orbitals and N-pz orbitals (Figure 4b), confirming the strong bond between Fe and N atoms. We also concluded the semiconductor character for Fe-NH MOF since there are no states at the Fermi level. For the Fe-O MOF in Figure 4c, the metallic feature mainly comes from the contributions of Fe-dxy, Fe-dx^2^-y^2^, O-px, and O-pz orbitals, which also mainly give the spin magnetism and hybridizations of the Fe-O bond. For Fe-S MOF in Figure 4d, S-px, S-pz, Fe-px and Fe-dx^2^-y^2^ existing at the Fermi level, these orbitals endow their metallicity, spin magnetism, and strong bond to Fe-S. 

### 2.3. HER

The hydrogen adsorption free energies are estimated under various configurations in this part to assess the HER activity of these Fe-MOFs. Four representative adsorption sites are chosen to show the HER catalysis activity, which are Fe, N/O/S, C_1_, C_2_, and C_3_ atoms in Figure 3. The corresponding adsorption structures are displayed in Appendix A. The calculated HER free energy diagrams of three Fe-MOFs at a potential *U* = 0 relative to the standard hydrogen electrode at pH = 0 are plotted in Figure 5. For Fe-NH MOF in Figure 5a, the Gibbs free energies of hydrogen adsorption (Δ*G*_H_) on Fe and N sites are 0.16 eV and 0.45 eV, while the Δ*G*_H_ is more than 0.88 eV on C sites, suggesting that the optimized HER activity is 0.16 eV. The Δ*G*_H_ of Fe-O MOF on Fe and O atoms are 0.60 eV and 0.08 eV, respectively, the C atoms also exhibit a poor HER catalysis performance and the Δ*G*_H_ > 0.67 eV. Similarly, the Fe-S MOF monolayer possesses poor HER catalytic behavior due to the high hydrogen adsorption free energies (Δ*G*_H_ > 0.37 eV). 

The reason can be deduced from the Bader charge analysis in Table 1, the O gains more electrons (+1.07 *e*) showing higher catalysis activity, the S and N gain fewer electrons (+0.13–+0.83 *e*) displaying poor catalysis properties. In comparison with the findings of previous studies (Table 2), the Fe-O MOF displays the small or comparable hydrogen adsorption free energy (Δ*G*_H_ = 0.08 eV), implying its excellent HER electrocatalytic activity. 

### 2.4. OER

The OER electrocatalytic activity of these three Fe-MOF monolayers was then evaluated. According to previous work, the OER process should consist of four elementary steps [37]. They are (1) *+ H_2_O → *OH+ H^+^, (2) *OH → *O + H^+^, (3) *O + H_2_O → *OOH + H^+^, and (4) *OOH → *+ O_2_ + H^+^, respectively. The optimized atomic structures of *OH, *O, and *OOH intermediates on three Fe-MOF monolayers are diagnosed by considering different adsorbed sites and conformations (Appendix A). It is found that their reaction sites are the same as Fe atoms (Figure 6), which is in agreement with those of other MOF materials [22,24].

The OER free energy diagram at 0.00 V, 1.23 V, and work potential is depicted in Figure 6. The OER free energy diagram of Fe-NH MOF plotted in Figure 6a indicates that the third step possesses the biggest uphill, and the elementary step *O + H_2_O → *OOH + H^+^ is the rate-limiting step. When the electrode is 2.15 V, all four-element steps are downhill. Thus, the calculated OER overpotential (*η*^OER^) is 0.92 V. The working potentials of Fe-O MOF and Fe-S MOF are 2.23 V and 2.45 V, where it can be deduced that their OER overpotentials are 1.00 V and 1.22 V. Therefore, the Fe-NH MOF exhibits the best OER catalytic activity for converting H_2_O to O_2_ in this study. We further compare the OER performance of Fe-MOF with the recent catalysts in Table 2. It is worth noting that the OER overpotential of Fe MOF (*η*^OER^ = 0.92–1.22 V) is 2–3 times that of the best-known OER catalyst IrO_2_ (0.45–0.59 V) [31], implying their poor OER electrocatalytic activity. The other methods should be adopted to tune the OER electrocatalytic activity of 2D Fe MOF monolayer. 

### 2.5. ORR 

The ORR electrocatalytic activity of the Fe–MOF sheets is discussed in this section. Previous studies have shown that the O_2_ dissociative pathway is difficult to achieve on 2D MOF materials, which are similar to Pt(111) [38] and several single-atom catalysts [28,36,39,40]. Here, the ORR can be regarded as the inverse process of OER, which is also described by four elementary steps (1) * + O_2_ + H^+^ → *OOH, (2) *OOH + H^+^ → *O + H_2_O, (3) *O + H^+^ → *OH, and (4) *OH + H^+^ → *+ H_2_O, respectively. The optimized atomic structures of ORR intermediates on three Fe-MOF monolayers are the same as OER intermediates on them and the active sites are Fe atoms (Appendix A). 

The calculated ORR free energy diagrams of three Fe-MOFs are depicted in Figure 7. The blue, black, and red lines represent the electrode potentials at 0.00 V, 1.23 V, and working potential. As displayed in Figure 7a, the elementary steps of Fe-NH MOF at 0.00 V are downhill. When the electrode potential is up to 0.85 V (which is defined as working potential, *U*_work_), the first step is spontaneous, thus the * + O_2_+ H^+^ → *OOH step is the rate-limiting step, which is defined as working potential. Thus, the corresponding ORR overpotential is 0.38 V, which can be calculated by the equation (*η*^ORR^ = 1.23–0.85). Similarly, it is worth noting from Figure 7b,c, that the rate-limiting steps on Fe-O MOF and Fe-S MOF are the first and the fourth steps. Their corresponding working potentials (*U*_work_) are 0.38 V and 0.48 V for Fe-O MOF and Fe-S MOF, indicating that their ORR overpotentials (*η*^ORR^) are 0.85 V and 0.75 V. Therefore, the Fe-NH MOF possesses the lowest overpotential, and the *η*^ORR^ is even lower than the best ORR catalyst of Pt (0.45 V) [38]. We also compare this *η*^ORR^ value with that of other excellent ORR catalysts (Table 2), which suggests that the Fe-NH MOF could boost ORR electrocatalysis performance.

To gain further insight into the catalysis property of Fe-MOF, we analyze the adsorption free energies of OOH, O, and OH species. Here we give the values of an ideal ORR catalyst, which are 3.69 eV for Δ*G**OOH, 2.46 eV for Δ*G**O, and 1.23 eV for Δ*G**OH, respectively. Furthermore, the Sabatier principle claims that an ideal catalyst should serve moderate adsorbed energies for all reaction species. For Fe-NH MOF, the values are Δ*G**OOH = 4.06 eV, Δ*G**O = 1.92 eV, and Δ*G**OH = 1.02 eV, the corresponding adsorbed energy differences are 0.10 eV (4.06–3.69), 0.54 eV (2.46–1.92), and 0.21 eV (1.23–1.02), respectively, which indicates that Fe-NH MOF exhibits strong adsorption to *O species. We also conclude that the adsorbed energy differences are 0.58 eV, 0.05 eV, and 0.06 eV for Fe-O MOF, and 0.16 eV, 0.80 eV, and 0.75 eV for Fe-S MOF, which suggests that Fe-O possesses weak adsorption energy to OOH, Fe-S displays strong adsorption strength to OH. These adsorbed energy differences could confirm the rate-limiting steps, which are the first step for Fe-NH and Fe-O MOFs, and the fourth step for Fe-S MOF. The results confirm the Sabatier principle [41]. In other words, the Fe atom in Fe-NH MOF loses 1.26 *e*, in Fe-O MOF it loses 1.51 *e*, and in Fe-S MOF it loses 0.58 *e* (Table 1), thus the active Fe exhibits different adsorption energies for various ORR species, which led to the different ORR catalysis activity. In summary, the coordinated ligand can promote the charge transfer between the central Fe atom and graphene nanosheets, and further regulate their chemical catalysis performance for HER, OER, and ORR. 

## 3. Methods

The first-principles calculations were performed with the context of spin-polarized DFT as implemented in the Vienna ab initio simulation package (VASP) [42]. The exchange-correlation approximation was described by the generalized gradient approximation with the Perdew–Burke–Ernzerhof function [43,44]. The plane-wave cutoff energy was 500 eV for the projected augmented wave approach [45]. The vacuum layer was larger than 15 Å. All geometry structures were allowed to fully relax until the Hellmann–Feynman force on atoms was less than 0.01 eV Å^−1^, and the total energy change was less than 1.0 × 10^−5^ eV. The grid density of k-point mesh in Monkhorst–Pack scheme was less than 2π × 0.01 Å^−1^. The ab initio molecular dynamics (AIMD) were conducted with the Nosé algorithm in the NVT ensemble to investigate thermodynamical stability [46]. To treat the exchange–correlation energy of the localized d-orbital of Fe atoms, the PBE+U calculations were employed by adding the Hubbard term to the Hamiltonian [29,47]. The DFT-D3 correction was adopted to describe the long-range van der Waals interaction [48]. The VASPKIT code was used to analyze the output files from the VASP. The free-energy change (Δ*G*) for each fundamental step was calculated by the equation [38],
(1)$$\Delta G=\Delta E+\Delta E_{\mathrm{ZPE}}-T\Delta S+\Delta G_{U}+\Delta G_{\mathrm{pH}}+\Delta G_{\mathrm{field}}$$
where Δ*E* is the electronic energy difference, Δ*E*_ZPE_ is the zero-point energy (ZPE), T is 298.15 K, and ΔS is the difference in entropy. Δ*G*_U_ = e*U*, where U is the electrode potential, and e is the electron transfer. Δ*G*_pH_ = *k*_B_T × ln10 × pH, where *k*_B_ is the Boltzmann constant, and pH = 0 in this study, which is the same as previous works [24,31,35,36]; Δ*G*_field_ is neglected. The entropy and vibrational frequencies of the gas species are taken from the database [49]. 

## 4. Conclusions

To summarize, we have obtained the atomic geometry, stability, HER, OER, and ORR electrocatalytic activity of Fe-MOF by using first-principles calculations. The Fe-MOFs with NH, O, and S ligands possess high stability and are atomically thin like 2D graphene. It was found that the coordinated ligands (NH, O, and S) could promote the charge redistribution in Fe-MOF, and further regulated their electronic structures and chemical performance. These three Fe-MOFs possess spin magnetism. The calculated free energy diagrams indicate that the Fe-O MOF displays the smallest hydrogen adsorption free energy (Δ*G*_H_ = 0.08 eV), the Fe-NH MOF exhibits the best ORR catalysis performance with the overpotential of 0.38 V. Unfortunately, these three Fe-MOFs possess poor OER properties due to the *η*^OER^ > 0.92 V. Our computational results offer not only a promising strategy for the design of high efficiency versatile electrocatalysts, but also promote the following experimental exploration on the use of 2D MOF in water splitting, fuel cells, and metal-air batteries. 

## Figures and Tables

**Figure 1 molecules-27-01528-f001:**
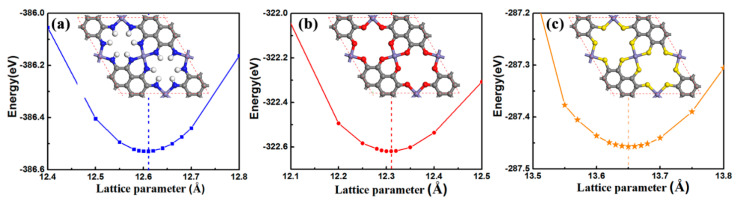
Optimized lattice parameters of (**a**) Fe-NH-MOF, (**b**) Fe-O-MOF, and (**c**) Fe-S-MOF monolayers.

**Figure 2 molecules-27-01528-f002:**
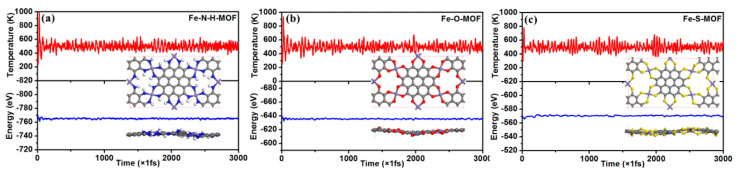
The AIMD simulations of (**a**) Fe-NH-MOF, (**b**) Fe-O-MOF, and (**c**) Fe-S-MOF monolayers at 500 K during the timescale of 3 ps.

**Figure 3 molecules-27-01528-f003:**
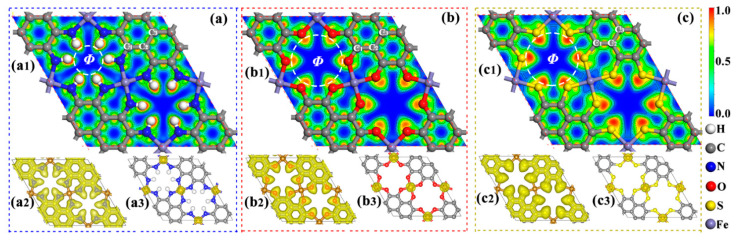
Geometries (1), electron localization function (ELF) (2), and spin density (3) of (**a**) Fe-NH-MOF, (**b**) Fe-O-MOF, and (**c**) Fe-S-MOF monolayers.

**Figure 4 molecules-27-01528-f004:**
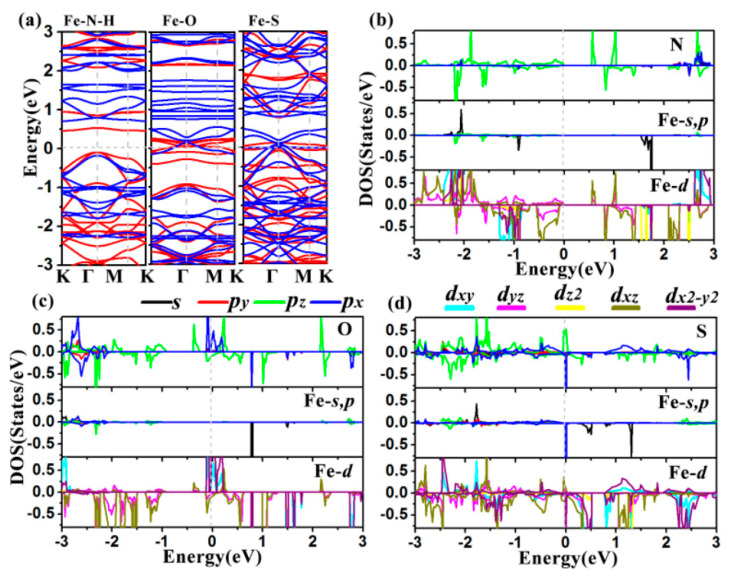
Band structures (**a**) and projected density of states (PDOS) of (**b**) Fe-N-H-MOF, (**c**) Fe-O-MOF, and (**d**) Fe-S-MOF monolayers. The Fermi levels (EF) are set to 0 eV.

**Figure 5 molecules-27-01528-f005:**
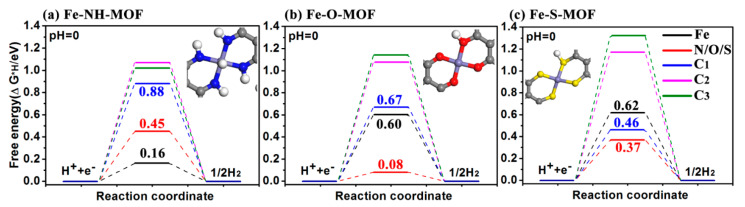
Calculated free energy diagram for hydrogen evolution on (**a**) Fe-NH-MOF, (**b**) Fe-O-MOF, and (**c**) Fe-S-MOF monolayers at a potential *U* = 0 relative to the standard hydrogen electrode at pH = 0. Insets are corresponding optimized adsorbed H intermediates.

**Figure 6 molecules-27-01528-f006:**
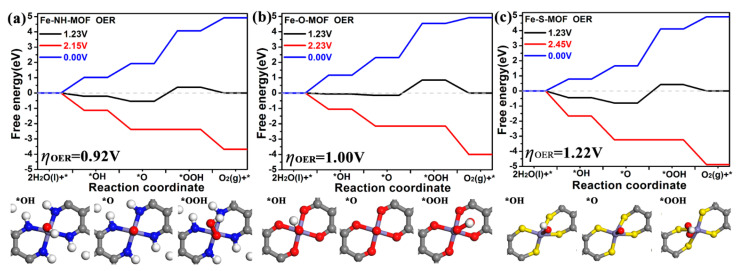
Calculated free energy diagram for OER on (**a**) Fe-NH-MOF, (**b**) Fe-O-MOF, and (**c**) Fe-S-MOF monolayers at a potential *U* = 0, 1.23, and working potentials relative to the standard hydrogen electrode at pH = 0. Insets are corresponding OER intermediates.

**Figure 7 molecules-27-01528-f007:**
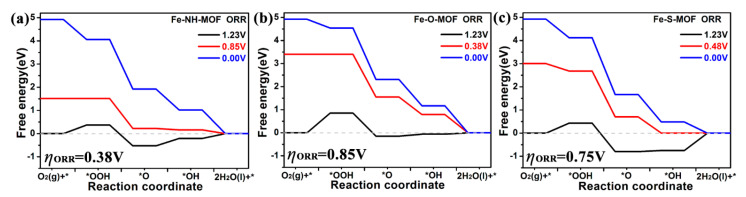
Calculated free energy diagram for hydrogen evolution on (**a**) Fe-NH-MOF, (**b**) Fe-O-MOF, and (**c**) Fe-S-MOF monolayers at a potential *U* = 0, 1.23, and working potentials relative to the standard hydrogen electrode at pH = 0.

**Table 1 molecules-27-01528-t001:** Calculated geometric parameters (lattice constants (*l*_a_), the bond length of Fe-N (O, S) (*D*_Fe-N (O,S)_), C-N (O,S) (*D*_C-N (O,S)_), and diameter of the hole (*Φ*) (Å)), total magnetic moment (*M*_tot_, *μ*_B_) and Bader charge (*Q*_Fe_ (*e*), *Q*_N,O,S_ (*e*)) of Fe-NH-MOF, Fe-O-MOF, and Fe-S-MOF monolayers.

Materials	*l*_a_ (Å)	*D*_Fe-N (O,S)_ (Å)	*D*_C-N (O,S)_ (Å)	*Φ* (Å)	*M*_tot_ (*μ*_B_)	*Q*_Fe_ (*e*)	*Q*_N,O,S_ (*e*)
Fe-NH-MOF	12.61	1.85	1.35	3.44	6.00	−1.26	+0.83
Fe-O-MOF	12.31	1.83	1.30	5.17	10.45	−1.51	+1.07
Fe-S-MOF	13.65	2.15	1.74	5.74	9.25	−0.58	+0.13

**Table 2 molecules-27-01528-t002:** Comparison of the HER (Δ*G*_H_, eV), OER (*η*^OER^, V), and ORR (*η*^ORR^, V) catalysis performance in our results with previous literatures.

Materials	Δ*G*_H_ (eV)	*η*^O^^ER^ (V)	*η*^O^^RR^ (V)	Materials	Δ*G*_H_ (eV)	*η*^O^^ER^ (V)	*η*^O^^RR^ (V)
Fe-NH-MOF	0.16	0.92	0.38	IrO_2_ [31]	-	0.45–0.59	-
Fe-O-MOF	0.08	1.00	0.85	Co-BP [32]	-	0.42	0.36
Fe-S-MOF	0.37	1.22	0.75	Ni-BP [32]	-	0.44	0.29
V-W_2_B_2_O_2_ [33]	0.01–0.15	-	-	Pt-BP [32]	-	0.25	0.32
ZnW_2_B_2_O_2_ [33]	0.14–0.26	-	-	Fe-BHT [24]	-	0.88	-
Zn@InSe [34]	0.02	-	-	Ir_3_(HITP)_2_ [35]	-	-	0.31
Ni@PR-GDY [36]	−0.05	0.29	0.38	Rh_3_(HITP)_2_ [35]	-	-	0.37

## Data Availability

Not applicable.

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
