# Peer review of "Theoretical Investigation on the Hydrogen Evolution, Oxygen Evolution, and Oxygen Reduction Reactions Performances of Two-Dimensional Metal-Organic Frameworks Fe3(C2X)12 (X = NH, O, S)"

_molecules, 2022, doi:10.3390/molecules27051528_

Round 1

Reviewer 1 Report

Much similarity is found from given reference, present data in new way

Zhen Feng, Yi Li, Yanan Tang, Weiguang Chen, Renyi Li, Yaqiang Ma, Xianqi Dai. "Two[1]dimensional halogen-substituted graphdiyne: first-principles investigation of mechanical, electronic, optical, and photocatalytic properties" , Journal of Materials Science, 2020

Comments from the Reviewers:
Reviewer: 1
Comments to the Author
Reviewers Comments
Title: Theoretical investigation on the hydrogen evolution, oxygen evolution, and oxygen reduction reactions performances of two-dimensional metal-organic frameworks Fe3(C2X)12 (X=NH, O, S)
Comments

  1. The manuscript is quite interesting. The authors should check and report the similarity index to ascertain the originality of the work and its title. I recommend a minor revision as per the following comments provided below.
  2. In Abstract, “Therefore, we believe that this study will serve as a guide for developing of 2D MOFs-based SACs for water splitting” mention the basis or outcomes of this work for this believe.
  3. Additional experimentation theory and characterizations in-depth discussions should be added.
  4. The manuscript, even though well written, yet, there are many instances of misused of words such “So on” in the introduction (line 33&61). Mention original work instead of using “So on”.
  5. To clearly understand the novelty of the work a detailed comparative table with previous literature accompanied with its discussion is needed.
  6. How many experiments were conducted if all vital parameters are kept constant? What will be the effect of different pH values? Why pH=0 is used? Detailed experimental design should be provided for all these aspects covered in the manuscript.
  7. Influence of 500K temperature may change the geometries and stability of grapheme π-electrons which should be further discussed.
  8. Bulk of references may decrease the worth of manuscript and need to be reduced.

Author Response

Dear editors and referees,

Thank you very much for the letter and the positive comments concerning our manuscript entitled “Theoretical investigation on the hydrogen evolution, oxygen evolution, and oxygen reduction reactions performances of two-dimensional metal-organic frameworks Fe3(C2X)12 (X=NH, O, S)” (ID: molecules-1598121). Those comments are very helpful for revising and improving our manuscript. Based on the comments and your requests, we have made careful and extensive modification to the original manuscript. These modifications are highlighted in blue in the main text. Attached are our response and revised manuscript.

Once again, thank you and the reviewers for the good suggestions.    

Best regards, 

Zhen Feng

School of Materials Science and Engineering, Henan Institute of Technology,

Xinxiang, Henan, China  

Tel: +86-13462360890  

Reviewer 2 Report

In this work, the authors used DFT calculations to explore the feasibility of three 2D MOF catalysts for the HER, OER and ORR processes. The stabilities, free energy diagrams and overpotentials were computed. The materials studied contain the previously experimentally synthesized coronene ligand structure with Fe sites and varied coordinating atoms (NH, O, S). This work seems to be a follow up of the previous work by some of the authors, where they investigated a similar material (Mo with O ligand) for the NRR. While the methods and results are clearly presented, I feel that the current manuscript is too brief and may not contain enough new insights to be considered for publication at this stage. I suggest that the authors expand the scope of their study either by broadening the scope of materials studied, clearly explaining the reasons for the difference in performance of the three catalysts, and/or providing new insights for designing new 2D MOF catalysts.

I have two comments regarding the computational methods: (1) the use of the PBE functional should be clearly justified, because it may not be the most accurate for describing strongly correlated materials with transition metal centers (for instance, a DFT+U correction is often used), and (2) the detailed procedures for the AIMD simulations should be described in the Methods section.

Finally, there are a few spelling and grammatical mistakes throughout the text that should be corrected.

Author Response

(The authors gave the same response as above.)

Round 2

Reviewer 2 Report

The authors have adequately addressed the reviewers' comments and the manuscript is now publishable.